# Microstructure and Mechanical Properties of Hybrid AZ91 Magnesium Matrix Composite with Ti and SiC Particles

**DOI:** 10.3390/ma15186301

**Published:** 2022-09-10

**Authors:** Katarzyna N. Braszczyńska-Malik

**Affiliations:** Faculty of Production Engineering and Materials Technology, Institute of Materials Engineering, Czestochowa University of Technology, 19 Armii Krajowej Ave., 42-200 Czestochowa, Poland; kacha@wip.pcz.pl or k.braszczynska-malik@pcz.pl

**Keywords:** metal matrix composite, hybrid, magnesium, AZ91 alloy, SiC, Ti, microstructure, mechanical properties

## Abstract

In this paper, a new hybrid metal matrix composite, named AZ91/(SiC + Ti)_p_, is presented. The commercial AZ91 magnesium alloy was chosen as the matrix. The composite was reinforced with both SiC and Ti particles. The investigated material was successfully fabricated using stir casting methods. Microstructure analyses were carried out by digital and scanning electron microscopy with an energy-dispersive X-ray spectrometer (SEM + EDX). Detailed investigations disclosed the presence (besides the reinforced particles) of primary dendrites of the α phase, α + γ eutectic and some part of discontinuous precipitates of the γ phase in the composite microstructure. The composite was characterised by uniform distribution of the Ti particles, whereas the SiC particles were revealed inside the primary dendrites of the α phase, on the Ti particles and in the interdendritic regions as a mixture with the α + γ eutectic. Both the tensile and compression strength as well as the yield strength of the composite were examined in both uniaxial tensile and compression tests at room temperature. The fabricated AZ91/(SiC + Ti)_p_ hybrid composite exhibited higher mechanical properties of all those investigated in comparison with the unreinforced AZ91 matrix alloy (cast in the same conditions). Additionally, analyses of the fracture surfaces of the AZ91/(SiC + Ti)_p_ hybrid composite carried out using scanning electron microscopy (SEM + EDX) were presented.

## 1. Introduction

Metal matrix composites (MMCs) fabricated mainly by casting or powder metallurgy methods have been developed for many years. The main factors influencing the microstructure and properties of these composites are specified schematically in Figure 1. Light metal alloys based on aluminium or magnesium are mostly used for composite matrices, although different matrices such as Cu, Fe, Ti etc. are also applied. The reinforcements are chiefly ceramic particles, short fibers or whiskers such as C_graphite_, SiC, TiC, Al_2_O_3_, TiN (or different carbides, oxides, nitrides) and fly ash microspheres, nanotubes, graphene or MAX phases [1,2,3,4,5,6,7,8,9,10,11,12,13,14,15,16,17,18,19]. It should also be noted that in the last decade, metal–metal composites were also fabricated. In these materials, metallic reinforcements characterised by a high melting point and a lack of (or very low) solubility in the matrix alloy are used, for example, Ni, Cu, Ti, Ti6Al4V or Inconel 718 particles [20,21,22,23,24,25,26,27,28]. The choice of components is significant in order to obtain specific complexes of composite properties. It is important for the composite production process due to the possibility (or lack) of wetting the reinforcement by the liquid matrix and also to obtain a favourable type of interfaces between the components [1,2,4,8,29,30,31].

In recent years, hybrid metal matrix composites have also been intensively designed and investigated. These materials consist of more than one reinforcement (Figure 2) although composites with two or more shapes of reinforcement (like particles and fibers, nanotubes or whiskers) or composites with different materials used as reinforcement, such as SiC + TiC, SiC + Al_2_O_3_, C_graphite_ + SiC, Al_2_O_3_ + MoS_2_, etc., can be distinguished [29,30,31,32,33,34,35,36,37,38,39,40,41,42,43,44,45].

Hybrid composites have very wide possibilities of design and fabrication, owing to the very large number of reinforcement combinations. On the other hand, they are more difficult to manufacture because of the different type of reinforcement necessary to be used in most often one fabrication process [46,47,48,49,50,51,52,53,54,55,56,57]. They are also characterised by a more complex microstructure than classical composites with one type of reinforcement. Analyses and descriptions of the influence of particular reinforcing phases on the mechanical or wear behaviour are also more complicated. For example, Pitchayyapillai et al. [32] reported that Al_2_O_3_ particles increased the tensile strength of the Al6061/Al_2_O_3_/MoS_2_ hybrid composite, whereas the MoS_2_ reinforcement reduced this feature, though both types of reinforcing phases increased the wear and friction resistance of the composite. On the other hand, hybrid aluminium matrix composites with SiC and graphite particles, described in works [37,43,51,53], had excellent tribological properties but the SiC particles decreased the ductility, while C_graphite_ particles reduced the whole mechanical capability of the final material [43]. Geng et al. [54] revealed that SiC_w_ improved the strength and ductility of a 2024 Al matrix alloy, whereas SiC_p_ increased the wear properties, coefficient of thermal expansion and the elastic modulus. It should also be added that a comparison between different hybrid metal matrix composites is very difficult due to the design and investigations of materials with, among others, various types, sizes or volume fractions of reinforcements. For these reasons, the fabrication and detailed analyses of metal hybrid composites are still necessary in order to describe their behaviours.

In this work, the microstructure and main mechanical properties of a new hybrid composite based on the AZ91 magnesium alloy with both ceramic and metallic particles, i.e., SiC and Ti (named AZ91/(SiC + Ti)_p_), are presented. It should be noted that both the types of particles selected for this study are characterised by very good wettability by molten magnesium and possibilities of forming coherent interfaces with the matrix. In the case of Ti particles, it should be considered that both metals (i.e., Mg and Ti) have a hexagonal closed packed structure (HCP), with a very low lattice misfit value equal to 0.08 in the main directions of basal planes. It is also possible to create interfaces with small lattice mismatches in [01$\bar{1}$0] the direction of both cells, which are also equal to 0.08 [25,28,31]. Additionally, Ti has practically negligible dissolution in magnesium and does not form with Mg intermetallic phases either. On the other hand, Kandoh et al. [58] revealed (by sessile drop method) that the true contact angle between Mg and Ti was equal to about 31° (at 1073 K for 180 s). Silicon carbide is also very highly wettable by liquid magnesium and exhibits a complete lack of reactivity with magnesium [3,9,31]. Silicon carbide can also form coherent interfaces with magnesium [17,19,31]. For SiC particles with the 6H polymorph, the relationships of crystallographic orientations with a matrix of the following types were determined [59,60]: [$\bar{2}$113]_Mg_//[10$\bar{1}$0]_SiC_, (10$\bar{1}$1)_Mg_//(0006)_SiC_, ($\bar{2}$20$\bar{2}$)_Mg_//(1$\bar{2}$1$\bar{6}$)_SiC_ and [1$\bar{1}$00]_Mg_//[01$\bar{1}$0]_SiC_, (0002)_Mg_//(0006)_SiC_, (11$\bar{2}$0)_Mg_//($\bar{2}$110)_SiC_. For the reasons presented above, both types of particles may also play a role as advantageous places for the heterogeneous nucleation of magnesium [31]. The above attributes of the selected components allowed a composite to be obtained by the simple and inexpensive stir casting method. The main aim of this study was to analyse the microstructure and main properties of the fabricated hybrid composite.

## 2. Materials and Methods

The new experimental hybrid composite AZ91/(SiC + Ti)_p_ was fabricated on the basis of the commercial AZ91 magnesium alloy with the chemical composition given in Table 1. A mixture of titanium and silicon carbide powders was used as the reinforcement. Ti particles with the chemical composition according to ASTM B-348 (Grade 1) had a spherical morphology and a fraction below 50 μm. SiC particles with an irregular shape and of the 6H polymorph type had a fraction up to 4 µm. Both the powders were mixed in a ball mill in a volume ratio 50:50. Mechanical mixing of the powders caused some of the SiC particles to be partially embedded into the Ti particles. Figure 3 shows the SEM micrograph of the used powder mixture. A composite with a 15 vol.% reinforcing particle mixture was fabricated by the stir casting method, which consists of the mechanical mixing (under a protective atmosphere) of the molten matrix alloy with the reinforcing particles (adding during mixing). The prepared composite suspension was gravity cast into a cold steel mould in the form of rods 1.7 cm in diameter and 18 cm in height. In the same conditions, an unreinforced AZ91 matrix alloy was also cast for comparison.

For the microstructure analyses, specimens from the fabricated material were prepared by standard metallographic procedures with etching in a 1% solution of HNO_3_ in C_2_H_5_OH. The composite microstructures were observed with a Keyence VHS-7000 digital microscope (Keyence Corp., Osaka, Japan) and a JOEL JSM-6610LV scanning electron microscope (SEM) (JOEL Ltd., Tokyo, Japan) with an energy-dispersive X-ray spectrometer (EDX).

Tests of the composite mechanical properties were carried out on a Zwick/Roell Z100 machine (Zwick Roell Group, Ulm, Germany) with a strain rate of 0.01 mm/s, according to relevant ASTM standards. The ultimate tensile strength (UTS) and yield strength (TYS) were determined on standard rod-like samples with a diameter of 8 mm in a uniaxial tensile test. In the uniaxial compression test compression strength (CS) and yield strength under compression (YS) were also determined. In this investigation, specimens with a diameter of 8 mm and length of 12 mm were used. Uniaxial tensile and compression tests were carried out at room temperature. For comparison, the same mechanical tests were performed on the used unreinforced AZ91 magnesium matrix alloy (cast in the same conditions in the same mould as the fabricated composite). For each material, three specimens were tested. In addition, the fracture surfaces of the investigated composite after uniaxial tensile testing were also analysed by a JEOL JSM-6610LV scanning electron microscope (SEM) (JEOL Ltd., Tokyo, Japan) with an energy-dispersive X-ray spectrometer (EDX).

## 3. Results and Discussion

The AZ91 alloy used in this study as the matrix is a well-known and popular cast magnesium alloy from the magnesium–aluminium system. In the as-cast condition, the AZ91 alloy consists mainly of dendrites of the α solid solution (aluminium and zinc in magnesium) and the α + γ semi-divorced eutectic in the interdendritic regions (where γ is the Mg_17_(Al,Zn)_12_ intermetallic phase). The zinc present in the chemical composition of the alloy does not create separate phases but becomes built in the α and especially γ phases. This alloy exhibits very strong segregation of the alloying elements during solidification in a metal mould. It results in variable distribution of the alloying elements on the dendrite sections. An SEM micrograph with the EDX results of the unreinforced AZ91 matrix alloy illustrating changes in the chemical composition in different marked points of the microstructure is presented in Figure 4.

Figure 5 and Figure 6 show the typical microstructure observed in the fabricated AZ91/(SiC + Ti)_p_ hybrid composite. Both the types of used particles (i.e., SiC and Ti) were successfully introduced into the matrix alloy. It should also be noted that the investigated composite was characterised by uniform distribution of the Ti particles in the matrix. Analogical distributions of Ti particles were also observed in the magnesium matrix composites with only titanium particles (cast under comparable conditions) presented in previous studies [19,25,28,31]. Nevertheless, in the investigated composite microstructure, different distribution was revealed in the case of the silicon carbide particles. Some of them were observed inside the dendrites of the primary α phase, others on the titanium particles, and some were seen in the interdendritic regions.

Due to the very small size of the used SiC particles, their distribution was analysed in detail by the SEM + EDX method. Figure 7 presents the surface distribution of elements (Mg, Ti, Si and Al) on the observed microstructure, while Figure 8 shows the EDX results in the form of X-ray spectrums from designated points marked on relevant microstructure micrographs. It should be added that in the case of magnesium alloys (and composites), the electron beam penetrating the analysed areas during SEM + EDX investigations is rather high. For this reason, in the obtained results, for example, from the very small SiC particles themselves, the presence of other alloying elements (especially Mg) was also revealed. The results displayed in Figure 7b and Figure 8b,c unequivocally confirm the presence of silicon carbide particles inside the α + γ semi-divorced eutectic occurring in the interdendritic regions (for example: points 4–6 in Figure 8b and points 4–6 in Figure 8c). These figures also clearly show the presence of SiC particles inside the primary α phase (for example point 1 in Figure 8b and point 3 in Figure 8c). Additionally, Figure 7b also shows SiC particles present on the titanium particles. It should be noted that the revealed diversified distribution of SiC particles was not observed in the magnesium matrix composites with only silicon carbide particles (cast under comparable conditions), presented in previous studies [3,17,19,31]. On the other hand, Deng et al. [57] described a different distribution (interior grains and at the grain boundary) of SiC particles of two sizes (submicron and micron) in the AZ91 magnesium matrix composite after stir casting and hot deformation.

The presence of SiC particles on the Ti particles observed in the microstructure of the AZ91/(SiC + Ti)_p_ hybrid composite was largely due to the SiC embedded in the Ti during the powder mixture preparation process (Figure 3). On the other hand, the presence of SiC particles inside the α dendrites indicated that they could be favourable places for heterogeneous nucleation of the primary α phase. Nevertheless, it should be added that during solidification of the investigated AZ91/(SiC + Ti)_p_ hybrid composite, both types of particles can be advantageous places for heterogeneous nucleation of the primary α phase. Both types of particles exhibit very good wettability by molten magnesium, and both can create coherent interfaces with the α phase, which was described in the Introduction. Therefore, during solidification of the AZ91/(SiC + Ti)_p_ hybrid composite, the number of potential favourable places for heterogeneous nucleation was very high. It is very difficult to clearly determine which kind of particles were the main places of heterogeneous nucleation of the α phase, especially since the growth of α dendrites in magnesium alloys proceeds at an angle of 60°, which makes it difficult to unequivocally analyse the microstructure.

The detailed microstructure analysis also revealed that some parts of the silicon carbide particles were located in the interdendritic regions. These SiC particles were very often observed as a mixture with the α + γ semi-divorced eutectic, which is clearly visible in Figure 5b, Figure 6b, Figure 7b and Figure 8b,c. It is well known that good wettability particles should be absorbed by growing dendrites. On the other hand, the phenomenon of the particles being pushed out by the solidification front occurs rather in the absence of or very poor wettability. In the investigated AZ91/(SiC + Ti)_p_ hybrid composite the pushing of some parts of the silicon carbide particles into the interdendritic regions occurred rather as a result of very rapid growth of the dendrites of the primary α phase during fast solidification in the cold metal mould. The distribution of SiC in the α + γ eutectic is associated with the distribution of the alloying elements, which were also pushed out by the growing primary dendrites of the α phase (depleted in Al or Zn in comparison to the equilibrium system; Figure 3).

Additionally, it should be taken into consideration that the thermal conductivity of the silicon carbide is less than that of magnesium. During fast solidification in the metal mould, some part of the SiC particles could have a higher temperature than the growing primary dendrites of the α phase and for this reason they were pushed into areas of the liquid phase with a local higher temperature. The observed distribution of some part of the SiC particles in the interdendritic regions can be called microsegregation. On the other hand, it should also be noted that evidence of the phenomena of macrosegregation such as sedimentation or floating of the used particle mixture was not observed due to the very good wettability of SiC and Ti by the molten magnesium matrix alloy. The same uniform distributions (in the macroscale) of the Ti or SiC particles on the cross-section of composite casts were also observed in the magnesium matrix composites with only one type of particles [28,31]. Additionally, as was to be expected, no chemical reactions between the components occurred during the composite fabrication process.

In the microstructure of the fabricated AZ91/(SiC + Ti)_p_ hybrid composite, evidence of the secondary precipitation process was also revealed. Microstructural analyses disclosed the presence of discontinuous precipitations of the γ phase (marked as γ_DP_ in Figure 5b, Figure 6b, Figure 7b and Figure 8c) distributed in the interdendritic regions and near eutectic areas, where the concentration of the alloying elements (i.e., Al and Zn) was the highest. These precipitates formed in the solid state below the solvus curve (determining the variable solubility of aluminium in the α phase). The γ_DP_ precipitates have a characteristic lamellar morphology of alternating plates of the γ secondary phase and near-equilibrium α matrix phase. This precipitate mechanism is characteristic for magnesium–aluminium alloys but is most often observed in casts made in sand moulds or initially heated metal moulds and after heat treatment consisting of solution annealing and aging. In the case of casts made in cold metal moulds, these precipitates are rather not observed due to the fast cooling of materials [31]. In the microstructure of the unreinforced AZ91 matrix alloy cast in the same conditions, discontinuous precipitates of the γ_DP_ phase were not observed either (Figure 4). On the other hand, these precipitates were, however, reported in the hybrid (SiC + TiC)_p_/AZ91 nanocomposite [29]. The presence of γ_DP_ precipitates in the microstructure of the investigated AZ91/(SiC + Ti)_p_ hybrid composite indicates that the reinforcing particle mixture slowed down the cooling of the investigated material.

The results of the mechanical tests performed on the fabricated AZ91/(SiC + Ti)_p_ hybrid composite are presented in Table 2 with analogical results obtained for the unreinforced AZ91 matrix alloy (cast in the same conditions). Figure 9a shows typical tension and compression curves recorded for the AZ91/(SiC + Ti)_p_ hybrid composite, whereas a comparison of the values obtained from the mechanical tests of the AZ91/(SiC + Ti)_p_ hybrid composite and the unreinforced matrix alloy is presented in Figure 9b. For the investigated composite, the ultimate tensile strength (UTS) and compression strength (CS) at room temperature were 141 and 351 MPa, respectively, while the yield strength (TYS) was 118 MPa and yield strength under compression (YS) was 175 MPa. The fabricated composite exhibited higher mechanical properties than the AZ91 matrix alloy in both uniaxial tensile and uniaxial compression tests. The largest difference in the properties between the composite and matrix alloy was observed in the yield strength (TYS), which was equal 25%. The AZ91/(SiC + Ti)_p_ hybrid composite exhibited a tensile strength (UTS) 17% higher than the AZ91 matrix alloy. The yield strength under compression (YS) and compression strength (CS) were 16% and 10% higher, respectively, for the composite than for the unreinforced matrix alloy. The obtained results are in agreement with those described for various magnesium matrix composites [10,17,23,25,31] in which various positive effects of individual SiC or Ti particles on the mechanical properties of the fabricated materials were also observed. On the other hand, for comparison, Guo et al. [29] described the mechanical properties of another hybrid—the (SiC + TiC)_p_/AZ91 nanocomposite in the as-cast state. They obtained a comparable level of ultimate tensile strength but a yield strength below 100 MPa.

Figure 10 presents SEM micrographs of typical tensile fracture surfaces of the investigated AZ91/(SiC + Ti)_p_ hybrid composite. These surfaces were characterised by river patterns and dimples, although magnesium alloys usually exhibit rather brittle through cleavage or quasi-cleavage fracture due to the hexagonal closed packed structure of magnesium. It should be noted that (during the uniaxial tensile test) the Ti particles did not undergo cracking. The same results were observed for different cast magnesium matrix composites with only Ti particles [25,31]. In contrast, the phenomenon of cracking through of SiC particles in magnesium matrix composites was observed in previous studies [17,31] due to the formation coherent interfaces between silicon carbide and magnesium. Nevertheless, in the case of the investigated AZ91/(SiC + Ti)_p_ hybrid composite, the size of the used SiC particles (below 4 µm) made unequivocal analyses impossible.

The results of detailed investigations of the fracture surfaces of the AZ91/(SiC + Ti)_p_ hybrid composite are presented in Figure 11. Higher magnifications of the SEM micrographs and added EDX results allowed the detection of SiC particles on the fracture surface. The energy-dispersive X-ray spectrometry (EDX) results confirmed the presence of SiC particles in the received micrograph; although they obtained from the fracture surfaces were burdened with errors (especially quantitatively). The micrographs shown in Figure 11 illustrate that the SiC particles were present in all areas of the fracture surface, i.e., on the Ti particles and in the matrix phases. On the other hand, the presented SEM micrographs also reveal that the cracking process did not occur precisely at the Ti and matrix interfaces but in the matrix phases because the Ti particles visible in Figure 11 remained covered by the matrix. An analogical phenomenon was observed in cast AME505-Ti_p_ composites [25]. It indicates that the Ti particles were strongly connected with the magnesium matrix and confirms the creation of coherent interfaces between these components. Additionally, the small dimples and river patterns visible on the Ti particles denote that in these areas cracking was proceeded by the plastic fracture mechanism.

The presented results of the investigation of the AZ91/(SiC + Ti)_p_ hybrid composite unequivocally indicate that this type of material can be successfully fabricated by the stir casting method. Analyses of the fabricated composite indicate the advantageous influence of both types of used particles on the mechanical properties of the AZ91 magnesium matrix alloy. The AZ91/(SiC + Ti)_p_ hybrid composite was characterised by higher values of the studied mechanical properties than the unreinforced matrix alloy cast in the same conditions. Although the obtained increase was not very high (of the order of 10÷25%), the fact that as-cast particulate metal matrix composites very often do not achieve high mechanical properties should be taken into account. The increase in these properties above the level characteristic for an unreinforced matrix alloy seems to be the most satisfactory result. The next planned investigations must concern, first of all, a detailed study of the tribological properties of this type of composite.

## 4. Conclusions

In the presented paper, a new hybrid composite designed on the basis of the AZ91 magnesium alloy with Ti and SiC particles was studied. The main conclusions drawn are as follows:
A new AZ91/(SiC + Ti)_p_ hybrid composite was successfully fabricated by the relatively simple and inexpensive stir casting method;The microstructure of the obtained composite was characterised by uniform distribution of the Ti particles and microsegregation of some part of the SiC particles, although the rest of them were enclosed inside primary dendrites of the α phase. The microstructure of this material (besides the presence of reinforcing particles) consisted of dendrites of the primary α phase, α + γ eutectic and γ_DP_ discontinuous precipitates;The fabricated hybrid composite exhibited higher mechanical properties than the unreinforced matrix alloy. The fracture surface observations revealed that during the uniaxial tensile test, the cracking process of the composite proceeded mainly through the matrix phases.

## Figures and Tables

**Figure 1 materials-15-06301-f001:**
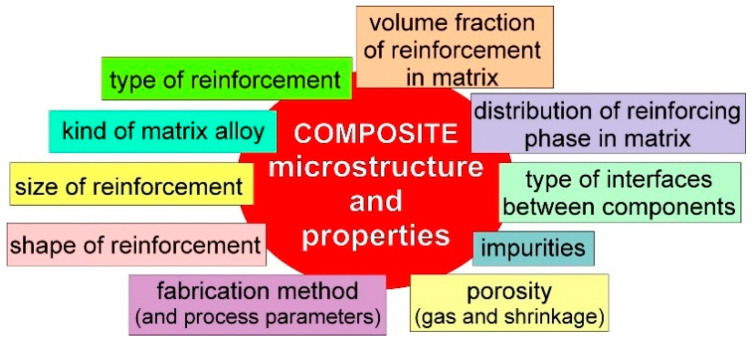
Scheme of factors influencing microstructure and properties of composites.

**Figure 2 materials-15-06301-f002:**
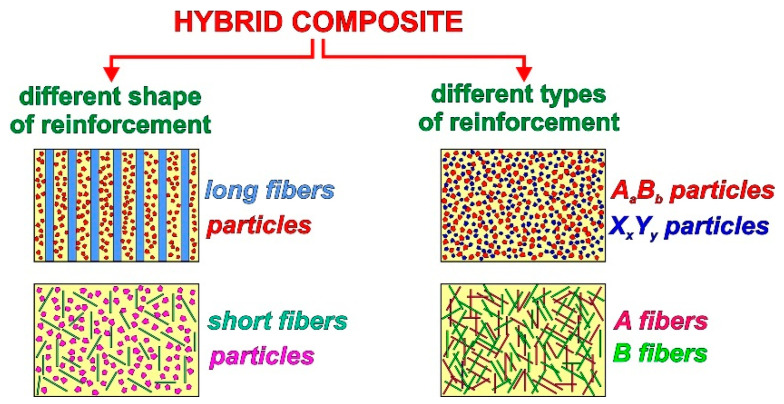
Scheme of various hybrid composites.

**Figure 3 materials-15-06301-f003:**
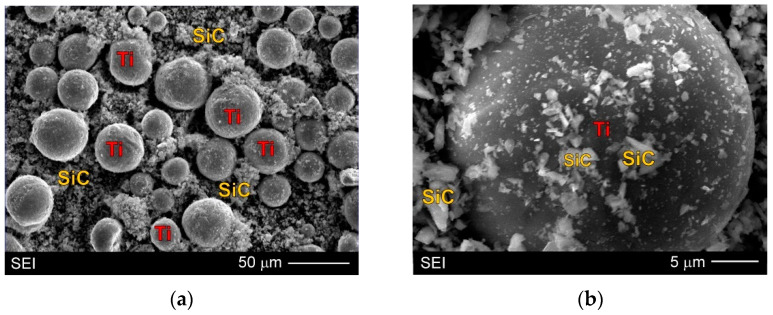
SEM micrographs of used Ti and SiC particle mixture: (**a,b**) Images taken at different magnification.

**Figure 4 materials-15-06301-f004:**
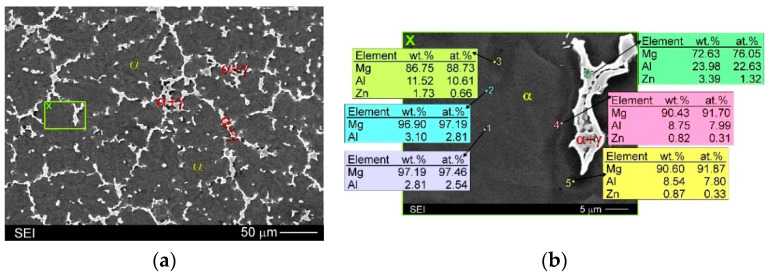
Microstructure of unreinforced AZ91 matrix alloy: (**a**) SEM micrograph; (**b**) Higher magnification of area marked as X on (**a**) micrograph with EDX results obtained from designated points.

**Figure 5 materials-15-06301-f005:**
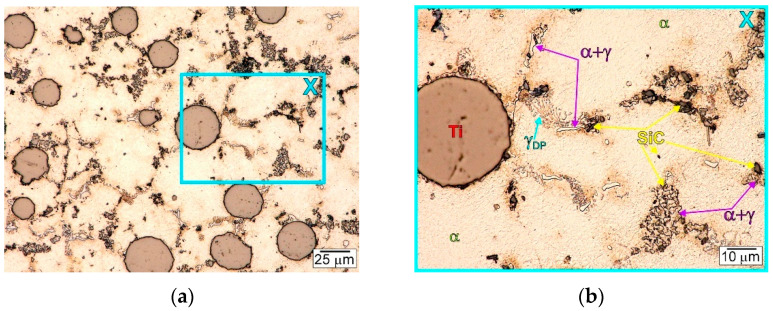
Microstructure of AZ91/(SiC + Ti)_p_ hybrid composite: (**a**) Digital microscope micrographs of etched surface; (**b**) Higher magnification of area marked as X on (**a**) micrograph.

**Figure 6 materials-15-06301-f006:**
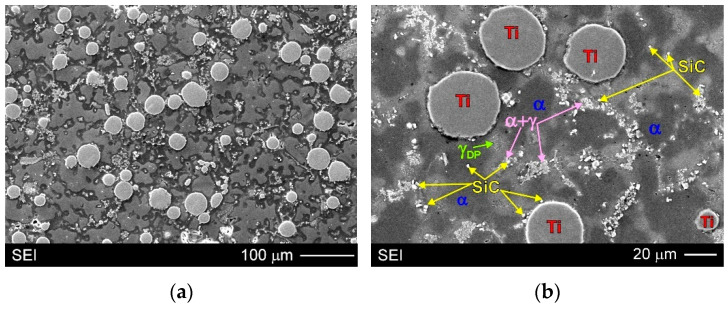
Microstructure of AZ91/(SiC + Ti)_p_ hybrid composite: (**a**,**b**) SEM micrographs taken at different magnifications of various areas of etched surface.

**Figure 7 materials-15-06301-f007:**
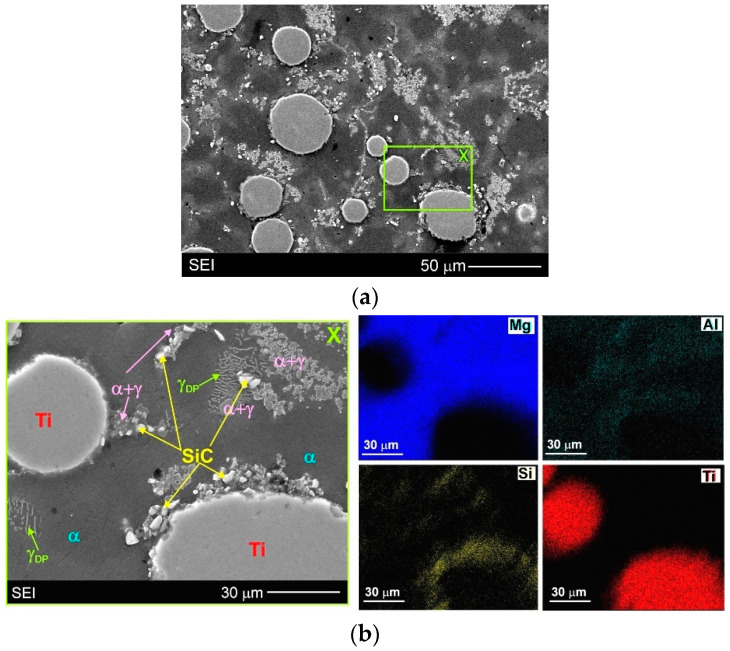
Microstructure of AZ91/(SiC + Ti)_p_ hybrid composite: (**a**) SEM micrograph; (**b**) Higher magnification of area marked as X on (**a**) micrograph with EDX results in form of surface distribution of main elements (magnesium, aluminium, silicon and titanium).

**Figure 8 materials-15-06301-f008:**
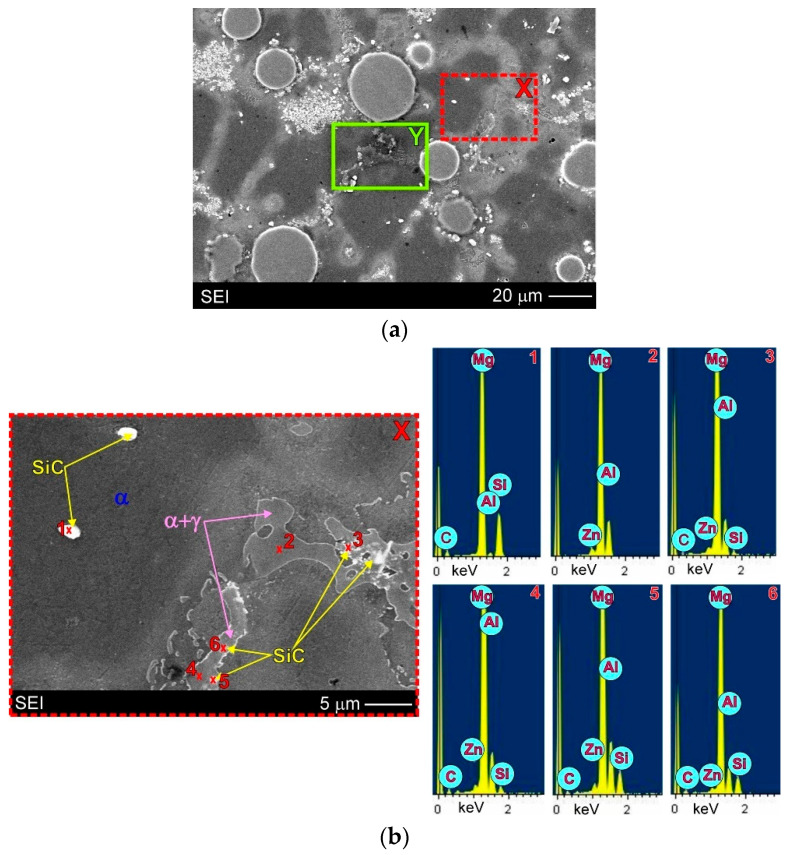
Microstructure of AZ91/(SiC + Ti)_p_ hybrid composite: (**a**) SEM micrograph; (**b**) Higher magnification of area marked as X on (**a**) micrograph with EDX results in form of X-ray spectrums from designated points; (**c**) Higher magnification of area marked as Y on (**a**) micrograph with EDX results in form of X-ray spectrums from designated points.

**Figure 9 materials-15-06301-f009:**
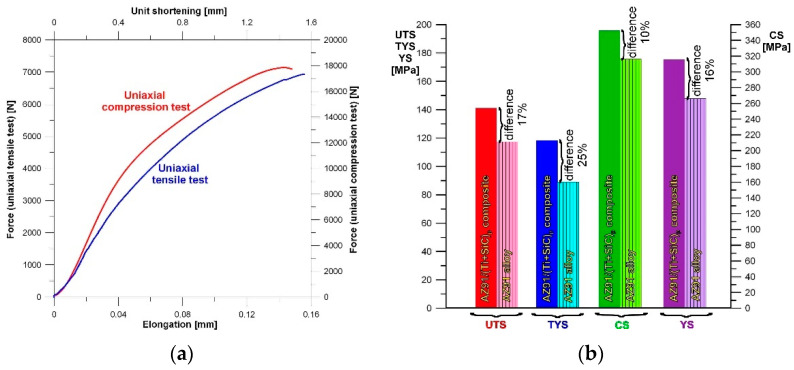
(**a**) Representative tension and compression curves for AZ91/(SiC + Ti)_p_ hybrid composite; (**b**) Ultimate tensile strength (UTS), yield strength (TYS), compression strength (CS) and yield strength under compression (YS) of AZ91/(SiC + Ti)_p_ hybrid composite compared with unreinforced AZ91 matrix alloy.

**Figure 10 materials-15-06301-f010:**
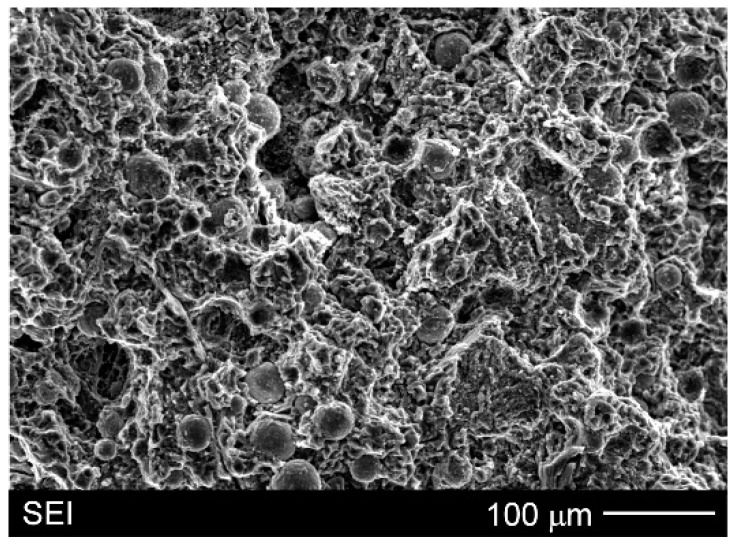
SEM micrograph of fracture surface of AZ91/(SiC + Ti)_p_ hybrid composite (after uniaxial tensile test).

**Figure 11 materials-15-06301-f011:**
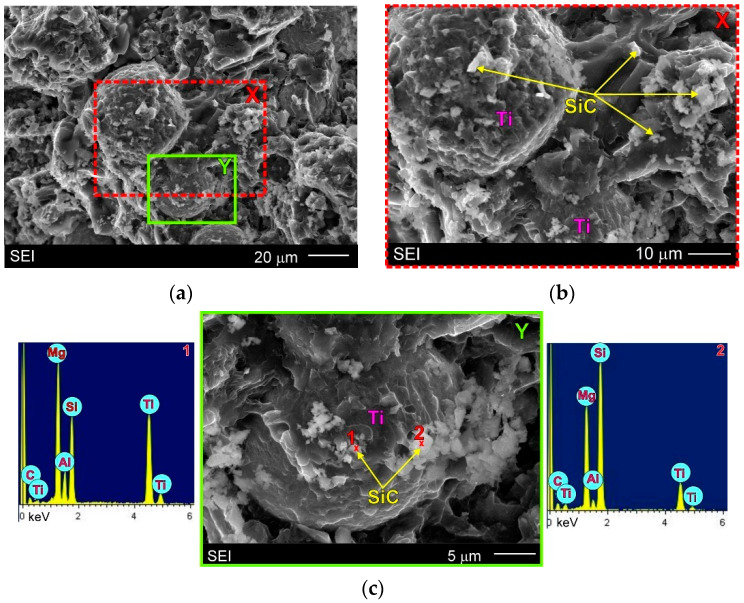
(**a**) SEM micrograph of fracture surface of AZ91/(SiC + Ti)_p_ hybrid composite (after uniaxial tensile test); (**b**) Higher magnification of area marked as X on (**a**) micrograph; (**c**) Higher magnification of area marked as Y on (**a**) micrograph with EDX results in form of X-ray spectrums from designated points.

**Table 1 materials-15-06301-t001:** Nominal chemical composition of used magnesium matrix alloy [31].

Alloy	Chemical Composition wt.%			
	Al	Zn	Mn	Mg
AZ91	8.7	0.7	0.13	rest

**Table 2 materials-15-06301-t002:** Ultimate tensile strength (UTS), yield strength (TYS), compression strength (CS) and yield strength under compression (YS) of AZ91/(SiC + Ti)_p_ hybrid composite compared with unreinforced AZ91 matrix alloy (cast in same conditions).

Material	UTS	TYS	CS	YS
AZ91/(Ti + SiC)_p_ composite	141	118	351	175
AZ91 matrix alloy	117	89	315	147

## Data Availability

Not applicable.

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
