# Peer review of "Microstructure and Mechanical Properties of Hybrid AZ91 Magnesium Matrix Composite with Ti and SiC Particles"

_materials, 2022, doi:10.3390/ma15186301_

Round 1

Reviewer 1 Report

The present research investigates fabrication of a new hybrid metal matrix composite, named AZ91/(SiC+Ti)p. The composite was reinforced with both SiC and Ti particles. The investigated material was successfully fabricated using stir casting methods. Microstructure analyses were carried out by digital and scanning electron microscopy with an energy dispersive X-ray spectrometer (SEM+EDX).

The present work represents excellent performed work in the field of Materials Science & mechanical Engineering and can be interested to the readers of this journal.

-         After careful revision of this paper, I advise to be accepted as the presented version. There is no comment except to accept this paper.

Author Response

Thank you very much for reviewing my manuscript.

Reviewer 2 Report

1. The star casting method is a method of liquid metallurgy in which the materials of the second phase are introduced into the molten matrix and allow the mixture to solidify. It is very important to ensure good wetting between the reinforcing elements and the melt. The author needs to provide data on the mutual wettability of the initial particles Ti and SiC with matrix melt.

2. What was the melt temperature and mixing time? It is necessary to describe in more detail the star casting process.

3. The process was carried out in an atmosphere of air. How did the author takes into account the possibility of oxidation during the process? According to the EDS data, there is completely no oxygen in the alloy. It is necessary to explain this result.

4. Why are the XRD analysis data not provided?

5. Figs. 4, 8 and 11. What is the accuracy and resolution of energy dispersive X-ray analysis? How confident is it possible to assert and indicate phases based only on the results of the EDS analysis? The author needs to explain their conclusions regarding the results of the EDS analysis.

6. The paper [Ref.11] (Braszczyńska-Malik, K.N. Types of component interfaces in metal matrix composites on the example of magnesium matrix 393 composites. Materials 2021, 14, 5182, 1-18) is dedicated to magnesium hybrid alloys (four different magnesium alloys) reinforced with SiC and Ti particles. What is the difference between this work and [Ref.11]? Only the composition of the Mg-alloy?

Author Response

Thank you very much for reviewing my manuscript. I appreciate valuable comments which helped me to improve the paper.

Reviewer: 1. The star casting method is a method of liquid metallurgy in which the materials of the second phase are introduced into the molten matrix and allow the mixture to solidify. It is very important to ensure good wetting between the reinforcing elements and the melt. The author needs to provide data on the mutual wettability of the initial particles Ti and SiC with matrix melt.”

Author: The required information about particles was presented in Section 1. “... Kandoh et al. [58] revealed (by sessile drop method) that the true contact angle between Mg and Ti was equal to about 31° (at 1073 K for 180 s).” “For SiC particles with the 6H polymorph, the relationships of crystallographic orientations with a matrix” was given.

Reviewer: “2. What was the melt temperature and mixing time? It is necessary to describe in more detail the star casting process.”

Author: All process parameters were selected experimentally. I do not consider it intentional to provide them. The work concerns the microstructure and properties of the new hybrid composite.

Reviewer: “3. The process was carried out in an atmosphere of air. How did the author takes into account the possibility of oxidation during the process? According to the EDS data, there is completely no oxygen in the alloy. It is necessary to explain this result.”

Author: The process (fabrication of composite suspension) was carried out in a protective atmosphere. This information was added. I can add that the casting process was carried out “from under the slag (oxides)”, which often forms on the surface of the liquid magnesium alloys.

Reviewer: “4. Why are the XRD analysis data not provided?”

Author: The XRD analysis does not add anything new to the work. Therefore, I believe that it is not essential.

Reviewer: “5. Figs. 4, 8 and 11. What is the accuracy and resolution of energy dispersive X-ray analysis? How confident is it possible to assert and indicate phases based only on the results of the EDS analysis? The author needs to explain their conclusions regarding the results of the EDS analysis.”

Author: Concerning Fig. 4: The phase composition of the AZ91 alloy is well-known. But results presented in Fig. 4b indicate (significant in this case) variable distribution of the alloying elements on the dendrite sections, because this alloy exhibits very strong segregation of the alloying elements during solidification, especially in cold metal moulds. Concerning Figs. 9 and 12 (previously Figs. 8 and 11): The presented EDX results indicate especially the SiC particles. In the case of the SiC particles: some of them were observed inside the dendrites of the primary a phase, others on the titanium particles, and some were seen in the interdendritic regions. SEM+EDX results conformed in which specific places of the microstructure the SiC is located. Of course, due to the width and depth of the electron beam (and very small size of the used SiC particles), the results contain a significant amount of background. But peaks derive from Si (Figs. 9b,c and 12c) prove in which places of the microstructure the SiC is located.

Reviewer: 6. The paper [Ref.11] (BraszczyÅ„ska-Malik, K.N. Types of component interfaces in metal matrix composites on the example of magnesium matrix 393 composites. Materials 2021, 14, 5182, 1-18) is dedicated to magnesium hybrid alloys (four different magnesium alloys) reinforced with SiC and Ti particles. What is the difference between this work and [Ref.11]? Only the composition of the Mg-alloy?”

Author:  The earlier article quoted above does not apply to hybrid composites at all. The paper [Ref.11] described interfaces between components in (binary) composites with only SiC particles, only Ti particles and only aluminosilicate cenospheres (and also with various magnesium matrices). I do not understand the term “magnesium hybrid alloys”.

P.S. (Author): All corrections required by Reviewers are marked in green. Additionally, the corrections required by the Editor are marked in yellow.

Reviewer 3 Report

This paper is a study of the  Microstructure and mechanical properties of hybrid AZ91
magnesium matrix composite with Ti and SiC particles

The paper is well written, and the English is good.

1. In the “Introduction, please rephrase  It is also possible to create interfaces with 78
small lattice mismatches in [011 2] the direction of both cells.”, please review the lattice direction.

2. In the “ Results and discussion” part of  3.4. In Figures 6, 7, and 8, please explain the HV you use for the SEM image acquisition and EDX spectrum, pressure, and WD (working distance). In your EDX spectrum you have O2, please add the percentual quantity, and remade the composition.

3. Please add an extra XRD spectrum, to see the Miller Indices.

 I recommend major revision.

Author Response

Thank you very much for reviewing my manuscript. I appreciate valuable comments which helped me to improve the paper.

Reviewer: “1. In the “Introduction, please rephrase  „It is also possible to create interfaces with  small lattice mismatches in [011 2] the direction of both cells.”, please review the lattice direction.”

Author: There was indeed a mistake at this point. This fragment is corrected. I calculated it again. A very low lattice misfit value equal to 0.08 is in the main directions and [01-10] the direction of basal planes.

Reviewer: “2. In the “ Results and discussion” part of  „3.4. In Figures 6, 7, and 8, please explain the HV you use for the SEM image acquisition and EDX spectrum, pressure, and WD (working distance). In your EDX spectrum you have O2, please add the percentual quantity, and remade the composition.”

Author:  Investigations were carried out in HV. Pressure was about 3x10-5 Pa. SEM images were obtained at 20kV, WD10mm, SS44. Of course, due to the width and depth of the electron beam (and very small size of the used SiC particles), the results (obtained for composite) contain a significant amount of background. For this reason I consider adding the results of quantitative analysis to be pointless. The presented EDX results indicate especially the SiC particles locations. Peaks derive from Si (Figs. 9b,c and 12c (previously Figs. 8 and 11)) prove in which specific places of the microstructure the SiC is located. In the case of the SiC particles: some of them were observed inside the dendrites of the primary a phase, others on the titanium particles, and some were seen in the interdendritic regions. The surface distribution of main elements (especially silicon) presented in Fig. 8b (previously Figs. 7b) also presents the SiC particles locations. In the case of magnesium matrix composites, oxygen is often present as a result of the preparation of the investigated surface (and etched).

Reviewer: “3. Please add an extra XRD spectrum, to see the Miller Indices.”

Author: The XRD analysis does not add anything new to the work. Therefore, I believe that it is not essential.

P.S. (Author): All corrections required by Reviewers are marked in green. Additionally, the corrections required by the Editor are marked in yellow.

Round 2

Reviewer 3 Report

The revised manuscript fulfills all the reviewer requirements.

I recommend to be accepted as it is.